# Angiopoietin1 Deficiency in Hepatocytes Affects the Growth of Colorectal Cancer Liver Metastases (CRCLM)

**DOI:** 10.3390/cancers12010035

**Published:** 2019-12-20

**Authors:** Nisreen S. Ibrahim, Anthoula Lazaris, Miran Rada, Stephanie K. Petrillo, Laurent Huck, Sabah Hussain, Shaida Ouladan, Zu-Hua Gao, Alexander Gregorieff, Rachid Essalmani, Nabil G. Seidah, Peter Metrakos

**Affiliations:** 1Department of Anatomy and Cell Biology, McGill University, Montreal, QC H3A 0G4, Canada; nisreen.ibrahim@mail.mcgill.ca (N.S.I.); miran.rada@mail.mcgill.ca (M.R.); 2Department of Surgery, McGill University Health Center Research Institute, Cancer Program, Montreal, QC H4A 3J1, Canada; anthoula.lazaris@mail.mcgill.ca (A.L.); stephanie.petrillo@muhc.mcgill.ca (S.K.P.); 3Departments of Critical Care and Medicine, McGill University Health Centre and Meakins-Christie Laboratories, Department of Medicine, McGill University, Montreal, QC 4A 3J1, Canada; laurent.huck@muhc.mcgill.ca (L.H.); sabah.hussain@mcgill.ca (S.H.); 4Department of Pathology, McGill University Health Center, Montreal, QC H4A 3J1, Canada; shaida.ouladan@mail.mcgill.ca (S.O.); zu-hua.gao@mcgill.ca (Z.-H.G.); alexander.gregorieff@mcgill.ca (A.G.); 5Laboratory of Biochemical Neuroendocrinology, Montreal Clinical Research Institute, University of Montreal, Montreal, QC H3T 1J4, Canada; rachid.essalmani@ircm.qc.ca (R.E.); nabil.Seidah@ircm.qc.ca (N.G.S.)

**Keywords:** angiopoietin, colorectal cancer liver metastases, co-option, histopathological growth patterns, angiogenesis

## Abstract

Colorectal cancer liver metastases (CRCLM) that receive their blood supply via vessel co-option are associated with a poor response to anti-angiogenic therapy. Angiopoietins (Ang1 and Ang2) with their Tyrosine-protein kinase receptor (Tie2) have been shown to support vessel co-option. We demonstrate significantly higher expression of Ang1 in hepatocytes adjacent to the tumor region of human chemonaïve and treated co-opting (replacement histopathological growth patterns: RHGP) tumors. To investigate the role of the host Ang1 expression, Ang1 knockout (KO) mice were injected intra-splenically with metastatic MC-38 colon cancer cells that develop co-opting liver metastases. We observed a reduction in the number of liver metastases and interestingly, for the first time, the development of angiogenic driven desmoplastic (DHGP) liver metastases. In addition, in-vitro, knockout of Ang1 in primary hepatocytes inhibited viability, migration and invasion ability of MC-38 cells. We also demonstrate that Ang 1 alone promotes the migration and growth of both human and mouse colon cancer cell lines These results provide evidence that high expression of Ang1 in the host liver is important to support vessel co-option (RHGP lesions) and when inhibited, favours the formation of angiogenic driven liver metastases (DHGP lesions).

## 1. Introduction

Colorectal cancer (CRC) is the second leading cause of cancer death in the Western world [1], most of which is due to metastatic disease [2]. A curative resection and neo-adjuvant chemotherapy (chemo) are associated with long-term survival and possible cure in a small group of CRCLM patients [3]. Currently, the combination of angiogenic inhibitors bevacizumab (AI, anti-VEGF, Bev) with chemo is used to improve resection rate and overall survival (OS) for CRCLM patients [3]. The response to this therapeutic strategy is mixed, with 10–15% of this group of patients being successfully converted to a resectable state. Unfortunately, 65–70% of the patients continue on chemotherapy until resistance develops and then may undergo up to four lines of treatment with an expected OS of 24–28 months [4]. We have no way of identifying those CRCLM patients that would respond/benefit to the addition of anti-angiogenic therapies (e.g. Bevacizumab). Recently, our group has identified that the response to Bev treatment is associated with the diversity of histopathological growth pattern (HGPs) within liver metastases (LM) [5]. Three HGPs have been identified: (1) the desmoplastic HGP, characterized by a desmoplastic stroma separating CRC cancer cells from the liver parenchyma; (2) the pushing HGP where the hepatic plates are pushed aside by the force of the growing tumor without invading the liver parenchyma and (3) the replacement HGP where tumor cells infiltrate the parenchymal cells in the liver as the lesions expand. The latter being associated with poor outcome to anti-angiogenic therapy [5] suggesting that the interaction between tumor and liver cells are equally important to the disease biology and response to therapy. Furthermore, the desmoplastic lesions have characteristics of hypoxia-driven angiogenesis, including increased fibrin deposition at the tumor-liver interface, increased endothelial cell proliferation, with associated inflammatory infiltrate at the tumor-stroma interface. In contrast, replacement HGP lesions grow by co-opting the sinusoidal blood vessels between the liver cell plates, without sprouting angiogenesis and with little perturbation of the liver architecture [6,7].

We have shown that patients with DHGP who received chemo plus Bev prior to resection had a significantly better pathologic response and OS than patients with RHGP [5]. Tumors can progress by hijacking pre-existing vessels using a co-option mechanism, which facilitates the migration of the tumor cells along the vessels of the host organ and the induction of angiogenesis through the stimulation of Ang1, Ang2 and Vascular Endothelial Growth Factor (VEGF) [8,9]. The co-opted vessels are usually supported by pericytes for stabilization, which is accompanied by the inhibition of endothelial cell (EC) proliferation and AI resistance [9,10]. Ang1 and Ang2 are vascular growth factor genes that have been identified as ligands for the receptor tyrosine kinase, Tie2 [11]. Ang1 is a highly soluble activating ligand and functions as a Tie2 agonist. It induces phosphorylation of Tie2 and activates downstream pathways that are important for ECs formation, survival, proliferation and migration [12]. Angl-Tie2 signaling protects against EC apoptosis and mediates vessel maturation by enhancing pericyte recruitment, inhibiting inflammation and VEGF-induced vessel permeability [13]. In various tumor types, aberrant overexpression of Ang1 remains controversial. Overexpression of Ang1 in breast and colorectal cancer cells delayed xenograft tumor growth [10,14]. In contrast, overexpression of Ang1 in glioblastoma cells resulted in extensive vasculature and accelerated tumor growth [15]. Expression of Ang1 promoted vascular normalization and perfusion, consequently improving the potency of chemotherapy in prostate xenograft [16]. However, inhibition of Ang1 expression reduced tumor angiogenesis, increased tumor cell apoptosis, decreased tumor necrosis and reduced tumor growth in HeLa cells [17]. Ang2 functions as an Ang1 antagonist that responds to pro-inflammatory stimuli and pro-angiogenic cytokines (e.g. VEGF), to promote angiogenesis by pericyte detachment and blood vessel destabilization [18]. In the absence of VEGF, Ang2 promotes EC apoptosis and consequent blood vessel regression in vivo [19]. Thus, activation of Ang2-Tie2 signaling in the presence of VEGF is necessary for tumor development and metastasis [20]. The role of angiopoietin proteins in CRCLM progression, angiogenesis and co-option is not well known. In this study, we characterized the vasculature of the DHGP and RHGP lesions, based on the expression of Ang1, Ang2 and Tie2 proteins and investigated the role of the host Ang1 expression in RHGP lesions using an Ang1 KO mouse model [21]. Our data demonstrated, for the first time, that the expression of Ang1 was increased in hepatocytes adjacent to tumor cells in chemonaïve and treated (chemo and chemo plus Bev) RHGP lesions. To investigate the importance of Ang1 expression in supporting the replacement phenotype, an in vivo mouse model demonstrated that loss of Ang1 expression in the host body diminished the number and size of metastasis and led to the formation of desmoplastic lesions. In addition, using an in vitro model where MC-38 cells were cultured with conditioned media from primary hepatocytes with an Ang1 knock out, we observed a decrease in cell viability, migration and invasion. The role of Ang1 was further validated in our in vitro experiments were the addition of recombinant Ang1 alone was sufficient to induce cell migration and growth of both human and mouse colon cancer cells. Thus, these results indicate the importance of the host Ang1 expression for sustaining the replacement HGP phenotype and tumor progression.

## 2. Results

### 2.1. Expression of Vascular Factors in Chemonaïve CRCLM Human Samples

We have previously shown that desmoplastic lesions contain immature vessels, whereas replacement lesions contain mature vessels, supporting the role of vessel co-option [22]. To investigate the vascular factors involved in co-option, we performed IHC staining on human CRCLM samples using antibodies to Ang1, Ang2 and Tie2. In order to correlate our previous vascular characterization using CD31, CD34//Ki67 and VEGFA with the current study, we used serial sections from the same samples used in our previous paper [22]. IHC staining was performed on a total of twenty-three chemonaïve lesions (DHGP: *n* = 11 and RHGP: *n* = 12). In chemonaïve RHGP lesions, we observed higher levels of Ang1 expression in the cytoplasm of hepatocytes adjacent to the tumor compared to the cytoplasm of tumor epithelial cells and hepatocytes distal to the tumor (Figure 1A–C). This increase was not observed in the DHGP lesions (Figure 1D–F). Positive staining was also observed in the blood vessel walls, as expected and thus served as an internal positive control (Figure 1B). We quantitated the levels of Ang1 staining and confirmed a significant increase of Ang1 positivity in adjacent normal hepatocytes compared to its distal normal and adjacent normal hepatocytes of DHGP lesion (*p*-value < 0.00005) (Figure 1G). We also observed a significant increase in the Ang1 positivity when comparing the central tumor to the peripheral tumor area of RHGP lesions (*p*-value < 0.005). This could be due to hepatocytes that have infiltrated the lesion (Figure 1G). When comparing the tumors of both RHGP and DHGP lesions, we observed a significantly higher level of Ang1 positivity in the RHGP tumor regions (CT and PT) than the DHGP tumor regions (CT vs PT: *p*-value < 0.005) (Figure 1G). Interestingly, in the DHGP lesion, the positivity of Ang1 in the adjacent normal hepatocytes was significantly higher than its distal normal (*p*-value < 0.005).

Since Ang1 is a secreted protein, we assessed if the Ang1 expression observed in the adjacent hepatocytes was not uptake but due to synthesis. We therefore performed FISH double staining using probes specific for Ang1 mRNA, followed by immunofluorescent staining for the cancer cell marker CK20. In a RHGP lesion, we observed a higher level of Ang1 mRNA in the hepatocytes at the interface region compared to the tumor region (Figure 2A). Furthermore, immunofluorescent co-staining of both hepatocyte specific antigen (HSA) and Ang1 protein demonstrated co-localization of HSA and Ang1 in the cytoplasm of hepatocytes adjacent to the tumor region (Figure 2B). In addition, staining with CD31 confirmed the presence of mature vessels next to the Ang1 expressing hepatocytes (Figure 2C).

We also examined the expression of Ang2. Ang2 was expressed in the ECs, cytoplasm of tumor epithelial cells at the periphery and center regions in the RHGP and DHGP lesions (Figure 3A,B). However, the positivity of Ang2 in the center region of the RHGP tumor was significantly higher compared with its periphery region and with the center region of the DHGP lesion (*p*-value < 0.005) (Figure 3C).

Another unique feature observed during our staining was the expression of Tie2. Tie2 was highly expressed in the ECs, cytoplasm of tumor epithelial cells at the periphery and center regions in the RHGP and DHGP lesions with no significant difference (Figure 3B,C). However, we observed high levels of positivity in immune cells in both lesion types (Appendix A). Our preliminary analysis shows the expression of Tie2 in leukocytes (CD45+), which are abundant around the DHGP ring (Appendix A) and randomly distributed in the adjacent normal of RHGP.

### 2.2. Expression of Ang1 in Treated (Chemo and Chemo Plus Bev) CRCLM Human Samples

It has been proposed that anti-VEGF (Bev) treatment normalizes tumor blood vessel structure by activating Ang1-Tie2 signaling [23]. Ang1 promoted vessel normalization in the tumor microenvironment by increasing pericyte coverage, reducing vascular leakiness and interstitial fluid pressure (IFP) resulting in improved blood flow and tumor perfusion in brain tumors [23]. We determined the expression of Ang1 in CRCLM after treatment with chemotherapy only and chemotherapy plus Bev. Ten lesions for each treatment groups were used, with a distribution of DHGP: *n* = 5 and RHGP: *n* = 5. These were serial sections from the same samples used in our previous paper, which indicated no difference in expression of VEGF in naïve vs treated samples [22]. However, in both chemo and chemo plus Bev treated RHGP lesions, the positivity of Ang1 remained high at the adjacent normal of the tumor, with no significant difference when compared to the chemonaïve samples (Appendix A). However, the expression of Ang1 was significantly up-regulated in the distal normal of the liver of chemo and chemo plus Bev samples compared to chemonaïve liver samples (*p*-value < 0.005) (Appendix A). Furthermore, RHGP lesions showed that the expression of Ang1 was significantly increased in the center of the tumor after treatment with chemo and not chemo plus Bev (*p*-value < 0.005) (Appendix A). No comparison with DHGP lesions is possible since the treated groups contain less than 10% viable tumor cells and cannot be evaluated [22].

### 2.3. Ang1 Deficiency Inhibits Liver Metastasis and Impacts HGP In Vivo

Since Ang1 was shown to be differentially expressed in the hepatocytes adjacent to replacement HGP lesions we asked whether host Ang1 deficiency could affect tumor growth or maintenance of the HGPs in a liver metastasis model or even possibly lead to the conversion of a replacement lesion into a desmoplastic lesion. To test this we used a conditional Ang1 KO mouse model [21] to perform intra-splenic injections of MC-38 cells. In control, non-induced mice (referred to as control mice), MC-38 cells developed replacement HGP lesions [22] that expressed Ang1, Ang2 and Tie2 (Appendix A). Furthermore, micrometastases can be identified in these lesions after Ang1 staining, as demonstrated by the increase in Ang1 expressing hepatocytes with increasing amounts of tumor cells (Appendix A). Following intra-splenic injections, a higher number of control mice (7/10, 70%) developed metastases with multiple large lesions (Figure 4A,B) compared to Ang1 KO mice (2/9, 22%). H&E staining showed that all the control mouse lesions formed replacement HGPs (Figure 4C) whereas, the Ang1 KO mice formed not only fewer and smaller metastatic foci but also desmoplastic lesions (Figure 4D).

To confirm that the Ang1 was indeed knocked out or knocked down, the expression of Ang1 in the host liver of control and Ang1 KO mice was assessed by IHC and qPCR (Figure 5A–C). In control mice, IHC staining demonstrated that the expression of Ang1 was significantly higher in the hepatocytes adjacent to the tumor compared to the hepatocytes distal to the tumor (Figure 5A). As expected, we observed low levels of Ang1 expression in the liver of the Ang1 KO mice compared to control mice (Figure 5B). The expression of Ang1 in control and Ang1 KO mice was also quantified by qPCR. The expression of Ang1 was significantly elevated in the control mice compared to Ang1 KO mice (*p*-value < 0.05) (Figure 5C). qPCR data was first normalized to b-actin and results presented as log2 fold change of control liver tissues with or without tumors for both groups relative to control livers for mice without injection.

Furthermore, we stained for CD31 to confirm that the control mice lesions had mature vessels and that the desmoplastic lesions formed in the Ang1 KO mice had less mature vessels, using angiogenesis, similar to what we observed in human lesions [22]. As shown in Figure 5, the number of mature blood vessels in the tumor of the control mice was higher (Figure 5D,F) compared to the number of blood vessel in the tumors from Ang1 KO mice (*p*-value < 0.0005) (Figure 5E,F), suggesting the presence of more immature vessels through vessel destabilization in the absence of Ang1.

### 2.4. Ang1 is Upregulated in Hepatocyte Cells Upon Co-Culture with MC-38 Colon Cancer Cells In-Vitro

To confirm the in-vivo findings that the interaction between tumor cells and hepatocytes leads to an increase in Ang1 in hepatocytes we developed an in-vitro system. Primary hepatocytes from Ang1 WT (*n* = 2) and Ang1 KO mice (*n* = 3) were isolated and cultured under different conditions (Appendix A). We first examined the percentage of Ang1 knock down in the hepatocytes harvested from the livers of mice that were induced to confirm the percentage of KO since this is an inducible system were doxycycline (DOX) is added to the drinking water and thus, we may not achieve 100% KO. Ang1 KO mice had approximately 60% reduction of Ang1 as shown by qPCR and western blot (Figure 6A,B). To test whether Ang1 expression in hepatocytes may be affected by the tumor cells interaction, Ang1 control and Ang1 KO primary hepatocytes were cultured with MC-38 cells using inserts to prevent contact, looking at secreted factors and also co-cultured to evaluate if any difference may be observed from conditioned media when the cells are in direct contact (Appendix A). As a first step we evaluated if we could observe up regulation of Ang1 in vitro in hepatocytes in the presence of colon cancer cells, when there is no direct contact (inserts experiment) but only exchange of media. Strikingly, the presence of MC-38 cells strongly increased the expression of Ang1 in the control hepatocytes compared to control hepatocytes cultured alone with only serum free medium, as demonstrated by western blot (Figure 6C, lane 1 vs 3). As expected, the Ang1 KO hepatocytes did not show this induction (Figure 6C, lane 2).

### 2.5. In-Vitro Effect of Ang1 Expression in Hepatocytes on MC-38 Cell Viability, Migration and Invasion

To further understand the cross talk between hepatocytes and the tumor cells, we assessed the effect of the Ang1 induction in hepatocytes, by the colon cancer cells, MC-38. We looked at cell viability and proliferation using the MTT assay, the migration of the cells using the scratch assay and finally the ability of cells to invade using the Boyden chamber assay. MC-38 cells were incubated for 24 h in conditioned media obtained from either Ang1 KO or control hepatocytes, cultured in serum free media with or without MC-38 cells (either with inserts or in direct contact: Appendix A). MC-38 cells incubated in serum free media served as a control for the experiment.

Significant inhibition of MC-38 cell viability was observed when they were incubated in conditioned media obtained from Ang1 KO hepatocytes cultured in direct contact with MC-38 cells (Figure 6D,E) compared to conditioned media obtained from Ang1 WT hepatocytes. However, no significant decrease in viability was found when MC-38 cells were cultured in conditioned media obtained from Ang1 KO hepatocytes cultured alone or Ang1 KO hepatocytes cultured with MC-38 using inserts, preventing cell-to-cell contact (data not shown). The invasion assay showed that the number of MC-38 cells that invaded the Matrigel and migrated through the pore in the membrane, was markedly lower when MC-38 cells were cultured in conditioned media obtained from Ang1 KO hepatocytes (Figure 7A,B). In addition, culturing MC-38 cells with conditioned media obtained from Ang1 KO hepatocytes dramatically reduced their migration abilities compared to culture with conditioned media from control hepatocytes (Figure 7C).

### 2.6. Effect of Recombinant Ang1 on Colon Cancer Cells Growth and Migration

To confirm that secreted Ang1 is responsible for the growth and migration of colon cancer cells we added recombinant Ang1 to one human colon cancer cell line (SW620) and mouse colon cancer cells (MC-38). Within 24 h of the addition of recombinant Ang1 to SW620 cells we observed a small, but significant increase in cell proliferation with the addition of 0.1uM Ang1, which was further increased with a lower dose of 0.05 μM Ang1 (Figure 8A). A similar effect with dosing was observed with MC-38 cells (Figure 8B). We then assessed the effect of the same two concentrations of recombinant Ang1 on cell migration using a scratch assay. As can be seen in Figure 8C,D, for SW620 and MC-38 cells respectively, migration of the cancer cells was increased with the addition of recombinant Ang1.

## 3. Discussion

Histopathological growth patterns of liver metastases have been shown to have distinct means of vascularization, which correlates with the patient OS. To characterize and evaluate the role of vascular factors in supporting these histopathological phenotypes, we performed IHC for the vascular markers Ang1, Ang2 and Tie2 on chemonaïve human CRCLM lesions. Our data demonstrated a distinct expression pattern of Ang1, Ang2 and Tie2 in the DHGP and RHGP chemonaïve human lesions. We observed significantly higher expression of Ang1 in the hepatocytes adjacent to the tumor in RHGP lesions compared to DHGP lesions. Furthermore, a marked infiltration of Ang1-expressing hepatocytes was observed in the tumor region of RHGP compared to DHGP lesions, as shown by co-localization of the hepatocyte marker HSA and Ang1 by immunofluorescence. FISH experiments showed that Ang1 mRNA was highly expressed in the adjacent hepatocytes compared to the tumor region of the RHGP lesion, thus the signal observed was de novo synthesis and not uptake of secreted Ang1. We therefore postulate that over expression of Ang1 by hepatocytes at the interface region of chemonaïve RHGP lesions may affect blood vessel formation at the edge of the tumor and induce blood vessel stabilization via paracrine effect. This is further supported by our findings on the expression of Ang2 or Tie2 in the tumor cells or hepatocytes of both RHGP and DHGP lesions. Tie2 expression was found significantly elevated in the adjacent normal and tumor regions (CT and PT) of RHGP lesions compared to DHGP lesions, consistent with the role of the strong agonist Ang1 in up-regulating Tie2 expression. Ang2 expression was shown in the tumor regions (CT and PT) and blood vessels in RHGP lesions.

Based on our present finding in chemonaïve RHGP lesions, overlapping signal between the expression of Ang1 and Tie2 along with CD31 and αSMA1 [22] suggest a role for high expression of the host Ang1 in stabilizing vessels in CRCLM with RHGP lesions. Therefore, vascular quiescence maintained by Ang1-Tie2 signaling found in RHGP lesions prevails over destabilization and pro-inflammatory Ang2-Tie2 signaling compared to DHGP lesions. Interestingly, the one difference in Tie2 expression observed was on the immune cells. The DHGP lesions have a denser region of leukocytes surrounding the lesion, which are Tie2 positive, compared to RHGP lesions which have a sparse distribution of leukocytes, also expressing Tie2. A number of studies have shown that presence of leukocytes is often accompanied with tumor angiogenesis [24]. In chemonaïve DHGP lesions, the presence of immature vessels [22], the abundance of leukocytes at the stromal ring and the expression of Tie2, Ang2 and VEGF [22] by tumor cells indicate activation of angiogenesis mechanism in this lesion as compared to the RHGP lesion. Further investigation into the type of leukocytes that express Tie2 is currently being performed.

In RHGP-treated CRCLM human samples, the expression of Ang1 remained high in the adjacent normal and significantly increased in the distal normal of the liver tissue of chemo and chemo plus Bev lesions compared to RHGP chemonaïve lesions. Low expression of Ang1 at the periphery of the chemo plus Bev RHGP lesions may correlate with inhibition of immature vessels normalization due to the Bev treatment that may be pruning immature vessels, leading to vessel stabilization. It has been reported that tumors with mature vessels were less sensitive to drug and acquired resistance after administration of angiogenic inhibitors [25], correlating with our findings.

To further understand the role of Ang1 in RHGP lesions, an inducible Ang1 KO mouse model was used. MC-38 cells that generate RHGP lesions were injected intra-splenically into both control and Ang1 KO mice. Our results demonstrated that metastasis, tumor growth and HGPs were strongly influenced by host Ang1 expression. We were able, for the first time, to demonstrate that knocking down of Ang1 in hepatocytes allowed for the formation of desmoplastic lesions. Furthermore, these lesions also presented with fewer mature vessels compared to the control replacement lesions. Therefore, inhibition of Ang1 expression in the liver inhibits mature blood vessel formation and hence tumor growth. We further demonstrated that the induced expression of Ang1 in hepatocytes is directly correlated with the presence of tumor cells, suggesting cross talk between the tumor and hepatocytes.

The data suggests that a RHGP lesion can be converted into a DHGP lesion and therefore questions if an established RHGP lesion can be converted to a DHGP. This is clinically significant, as we have shown that current treatment protocols (chemo + Bev) benefit patients with DHGP lesions, however we do no currently have any effective treatments for patients with RHGP lesions. If we could convert a RHGP lesion into a DHGP lesion then we can treat these patients and offer them improved outcomes. To address this control mice (*n* = 3) were injected intra-splenically with MC-38 colon cancer cells to develop tumors with RHGP lesions. These RHGP tumors were dissected into small fragments and then transplanted into control (*n* = 3) and Ang1 KO (*n* = 3) mouse livers. Control mice (*n* = 2) developed tumors with larger size (Appendix A) compared to Ang1 KO mouse (*n* = 1) which formed tumor with smaller size (Appendix A). H&E staining was then performed to assess the HGPs of these tumors. The tumors in the control mouse livers grew and developed a RHGP lesion (Appendix A). However, the tumor that grew in the Ang1 KO mouse formed a DHGP lesion (Appendix A). We have only performed these experiments on a few animals and are currently performing additional experiments. This study design will allow us to follow established liver metastases and then investigate if the phenotype can be reverted in a setting that would be more clinically significant. Furthermore the splenic injection model is also evaluating metastatic seeding and if we wish to validate a model for patient treatment regime having established single lesions will allow for a focused study.

Although we are confident as we have established desmoplastic lesions, we cannot account for the effects a whole body knockout may have on our findings, alternatively we can specifically target Ang1 with an Ang1-specific peptibody (ML4-3: Amgen, city, CA, USA). In future studies, the inhibitor can be injected alone or in combination with chemotherapy and/or anti-VEGF5 to assess if conversion to an angiogenic phenotype promotes drug sensitivity.

In-vitro, co-culturing primary control hepatocytes and MC-38 cells significantly induced the expression of Ang1 in the hepatocytes. It is possible that tumor cells that metastasize into the liver stimulate Ang1 expression in their environment (hepatocytes), which is important for blood vessel formation and growth of the tumor. It has been recently reported in human papillary thyroid carcinoma that upregulation of Ang1-Tie2 signalling activates pathways that are involved in cancer cells survival, proliferation and metastasis [26]. Furthermore, in-vitro, knockdown of Ang1 expression decreased the proliferation and migration of breast cancer cells, and found that the Ang1 overexpression rescued proliferation and migration of these cancer cells [27]. This is supported by our in-vitro data, where we demonstrate an increased migration and invasion of MC-38 cells cultured in control hepatocyte conditioned media and with the addition of recombinant Ang1. This phenotype was abolished when Ang1 was knocked out in hepatocytes, suggesting that secreted Ang1 from hepatocytes plays a critical role in tumor survival, migration and invasion and may have an independent function to its role in vascularization.

Turning to a clinical trial setting, the peptide– Fc fusion protein (AMG386/trebaninib) which blocks the binding of both Ang1 and Ang2 to Tie2, has struggled in three Phase III advanced ovarian cancer trials. However, our data suggests that not all lesions are driven by Ang1 and that stratification of patients into RHGP lesions and DHGP lesions, will allow for a better assessment of the efficacy of these drugs. With distinct differences between these two HGPs we believe that proper stratification of patients into Angiogenic Inhibitor responders (DHGP) and non-responders (RHGP) will permit a proper assessment of the efficacy of Angiogenic Inhibitors. Targeting treatment to patients that will respond to Angiogenic Inhibitors will: (1) increase downsizing of CRCLM in target populations, increasing the number of patients that can be converted from non-resectable to resectable, (2) increase 5 year overall survival in a subgroup of patients that will never be resectable and (3) reduce Angiogenic Inhibitor-associated morbidity in patients that do not benefit from and should not be treated with Angiogenic Inhibitors to improve their quality of life.

The next steps in moving closer to using this knowledge for the benefit to the patient is to identify biomarkers, preferably in the blood of patients, which can stratify these patients. Ang1 and Ang2 are normally present in blood serum and are generally found at equilibrium in a healthy individual. The blood serum ratio of Ang1 to Ang2 is quite low but elevates in inflammation, vascular regression, tumors and some diseases [28]. Based on our findings, we suggest that Ang1 can be used as a biomarker in CRCLM patients to predict response to anti-angiogenic therapy in the clinic. The identification/development of a blood biomarker may therefore be very feasible.

## 4. Materials and Methods

### 4.1. Clinical Data

This study included a total of 43 lesions from 43 patients. A prior written informed consent was obtained from all the subjects to participate in this study under a protocol approved by McGill University Health Centre Institutional Review Board (IRB: Study name: A Retrospective and Prospective, exploratory, translational study: Characterization of morphological and molecular biomarkers of tumor growth patterns in patients with liver metastases. #11-196-HGP). Clinical data was collected for each patient through the hospital database and medical records including demographics, primary and metastatic disease characteristics, relevant laboratory results, chemotherapy and co-morbidities. The median age of diagnosis was 63 (range 31–81) years. Rectal cancer accounted for 34% of the cases. Approximately two thirds (64%) of the patients had synchronous liver disease. Twenty-three lesions (11 DHGP and 12 RHGP) were chemonaïve, 10 lesions (5 DHGP and 5 RHGP) received chemo and 10 lesions (5 DHGP and 5 RHGP) received chemo plus Bev, with an average of seven cycles (Range 3–28). Estimated 1 and 3-year OS was 100% and 82.6% respectively. Twenty-seven (54%) patients had recurrence to the liver, estimated 1-year and 3-year disease free survival (DFS) was 49.9% and 44.4% respectively (26.5 months mean follow up duration).

### 4.2. Immuno-Histochemical (IHC) and Immunofluorescent (IF) Staining

Formalin-fixed paraffin-embedded (FFPE) human resected CRCLM and mouse liver with metastatic tumor blocks were used for this study. Serial sections 4 μm thick were cut from each FFPE block, mounted on charged glass slides (Superfrost Plus; Thermo Fisher Scientific, Hampton, NH, USA), baked at 650 °C for 1 h and then stored at 40 °C until use. For the IHC, sections were rehydrated and exposed to heat-induced epitope retrieval for 20 minutes in a citrate buffer (10 mmol/L citric acid, pH 6.0) using a steamer. Primary antibodies used include: Ang1 (ab102015, Abcam, Cambridge, UK,: 1:1500), Ang2 (ab153934, Abcam; 1:200), Tie2 (PA5-28582, Invitrogen, Carlsbad, CA, USA: 1:1250), human CD31 (M0823, Dako, Mississauga, ON, Canada: 1:20), mouse CD31 (DIA-310, Optistan, Cedarlane, Burlington, Ontario, Canada: 1:50), CD45 (MA5-13197, Dako: 1:500) and hepatocyte specific antigen (HSA, SC58693, Santa Cruz, Dallas, Texas, USA: 1:1500). The secondary antibodies were a goat anti-rabbit Alexa Fluor 488, goat anti-rabbit Alexa Fluor 594 conjugated antibodies (Molecular Probes, Life Technology, Carlsbad, CA, USA: 1:1000) using the Dako EnVision plus System-HRP Labelled Polymer Anti-Rabbit (cat. K4003) and Anti-Mouse (cat. K4007). All IHC slides were scanned at 40× magnification using the Aperio AT Turbo system. Images were viewed using the Aperio ImageScope ver.11.2.0.780 software program (Aperio Technologies Inc., Vista, CA, USA). Scoring and analysis was performed with the ImageScope software (algorithm: positive pixel count V9) [22]. Briefly, we randomly selected three areas for each of four different regions that were representing the central tumor (CT), periphery of tumor (PT), adjacent normal liver (ADN) and distal normal (DN) [22]. Then the algorithm calculated the positivity (Total number of positive pixels divided by total number of pixels: (NTotal – Nn)/(NTotal)) for each area. The average (3 areas) of each region was then used in our statistical analysis (Appendix A). Haematoxylin and Eosin (H&E) stained sections were prepared from all cases for an initial histopathological assessment with a pathologist. In the mouse model, CD31 IHC staining was used to count the number of blood vessels. We manually counted, using a 20x field, 10 regions per lesion and then the average was determined for each group.

### 4.3. Fluorescent in Situ Hybridization (FISH)

The RNAscope assay was performed manually using an RNAscope^®^ Multiplex Fluorescent Reagent Kit v2 according to the manufacturer’s (Advanced Cell Diagnostics USA, Newark, CA, USA) instructions. Briefly, the FFPE tissue sections were deparaffinized and pre-treated with hydrogen peroxide. Sections were incubated in RNAscope^®^ 1× Target Retrieval Reagent for 15 minutes at 99 °C and treated with RNAscope^®^ Protease Plus for 30 minutes at 40 °C in the HybEZ oven sequentially. Samples were incubated with RNAscope^®^ Probe-Hs-ANGPT1 482901 for 2 h at 40 °C. After three steps of amplification, RNAscope^®^ HRP-C1 was added on to the sections that were incubated after that with cyanine 3 for 30 min. RNA in situ hybridization was followed by IF protein staining for the pan-cytokeratin CK20 (ab76126, Abcam:1:250). Slides were mounted using mounting media (Fluoromount G 4958-02, Invitrogen) and confocal microscopy was performed using Carl Zeiss LSM 700 and Zen software (Zeiss International, Oberkochen, Germany).

### 4.4. Cell Culture, Mouse Experiments and Metastasis Induction

MC-38 mouse colon cancer cells form RHGP metastatic lesions in the liver of C57BL/6 mice [22]. MC-38 cells were grown in DMEM medium (Invitrogen) supplemented with 10% heat-inactivated fetal bovine serum (Hyclone, Logan, UT, USA) and 1% penicillin-streptomycin solution (450-201-EL, Wisent Bioproduct, St-Bruno, Qc, Canada). Inducible Ang1 KO mice were obtained from Dr. S. Quaggin [21]. Briefly, the ROSA-rtTA/ tet O-Cre transgenic system was used to generate inducible whole-body knockout of Ang1 upon administration of doxycycline (DOX) (Bishop, Canada) in drinking water. Floxed Ang1 males (129 background) were crossed with ROSA-rtTA (C57BL/6) females or with tet-O-Cre (C57BL/6) females. These F1 mixed mice (129/C57BL/6) were then interbred for three generations to generate the experimental Ang1fl/fl/Cre/Rosa-rtTA mice in a C57BL/6 background. The animals were kept on DOX in drinking water (2 mg/mL) for 4 weeks followed by one week on regular water [21]. Liver metastases in control (not treated with DOX) and Ang1 KO mice were generated by intra-splenic injection of 50 µL of DMEM media containing 2x105 viable MC-38 cells, followed by splenectomy 3 min after injection [29]. Control females (*n* = 10) and Ang1 KO females (*n* = 9) were used at 9–11 weeks of age in these experiments. Mice were killed 14–20 days post cells injection when animals in the control group became moribund. Visible metastases on the surface of the liver were enumerated and sized prior to fixation. Sections of the liver were also fixed in 10% buffered neutral formalin, and paraffin embedded. Experimental procedures were conducted in compliance with Canadian Committee on Animal Care guidelines in McGill University Health Center (MUHC, Montreal, QC, Canada).

Mice were genotyped by PCR using the following primer pairs; Ang1 flox (For 5′-CAATGCCAG AGGTTCTTGTGAA, Rev 5′-TCAAAGCAACATATCATGTGCA, Ang1 wt 233 bp, flox 328 bp), Ang1 del (For 5′-CAATGCCAGAGGTTCTTGTGAA, Rev 5′- TGTGAG CAAAACCCCTT TC, 481 bp), ROSA-rtTA (For 5′-GAGTTCTCTGCTGCCTCCTG, Rev 5′-A GCTCTAATGCGC TGTTAAT), general Cre allele (For 5′-ATGTCCAATTTACTGACCG, Rev 5′-CGCCGCATAA CCAAGTGAA, 673 bp) from Invitrogen.

### 4.5. Hepatocyte Isolation and Culture Conditions

Hepatocytes were isolated from adult control (*n* = 2) and Ang1 KO (*n* = 3) female mice using the two-step collagenase perfusion method as previously described [30]. Briefly, under anesthesia with 2% isoflurane inhalation, the peritoneal cavity was opened, and the liver was perfused in situ via the portal vein for 50 mL/10 min at 37 °C with calcium-free HEPES buffer I (142 mM NaCl, 6.7 mM KCl, 10 mM HEPES, pH 7.6) containing 0.19 mg/mL EGTA and for 50 mL/10 min, then followed by perfusion with calcium-supplemented HEPES buffer II (4.7 mM CaCl2, 66.7 mM NaCl, 6.7 mM KCl, 100 mM HEPES, pH 7.4) containing 0.5 mg/ml collagenase type V (Sigma Aldrich, Oakville, ON, Canada). To separate undigested tissue pieces, the suspended hepatocytes were passed through a 70 mm nylon filter into 50 mL Falcon tubes. The cell suspensions were centrifuged twice at 50 g for 5 min at 4 °C, and the cell pellet was resuspended in Williams’ medium E supplemented with 10% fetal bovine serum (Invitrogen) and 1% antibiotic-antimycotic mixture (15240-096, Gibco, Thermo Fisher Scientific). Cells were used only if cell viability was above 80% as assessed by trypan blue exclusion. Cells (1–1.5 × 10^6^) were seeded in Corning cell bind surface polystyrene 6-well plates (3335, Costar, Sigma Aldrich). After allowing for cell attachment for 2 h, cells were cultured in high glucose (25 mM) DMEM supplemented with 10% FBS and 1% antibiotic-antimycotic mixture overnight. The cells were then cultured in serum free media with/without MC-38 cells in direct contact (ratio 5 to 1) or using insert (353090, Falcon, Thermo Fisher Scientific) (Appendix A). After 24 h, the conditioned media from the different conditions were collected and frozen. The different conditioned media was added to MC-38 cells, incubated for 24 h and then the cells were analyzed.

### 4.6. RNA Extraction and qPCR

qPCR was performed from RNA extracted from the following mouse livers: wild type, non-induced mice (referred to as control mice) that received intra-splenic injections of MC-38 cells (*n* = 10); Ang1 KO mice that received intra-splenic injections of MC-38 cells (*n* = 9). Furthermore for the cell culture experiments RNA was extracted from isolated hepatocytes of Ang1 WT mouse livers (*n* = 2) and Ang1 KO mice livers (*n* = 3). Total RNA was extracted using the Qiagen (Hilden, Germany) RNeasy Plus Micro Kit (cat.# 7403), followed by reverse transcription (iScript reverse transcription supermix for qPCR kit 1708841, Bio-Rad (Hercules, CA, USA) and IQ Sybr Green supermix 1708882, Bio-Rad) according to the protocol of the manufacturer. TaqMan qPCR was performed with a MyiQ 2two color real time PCR system (Bio-Rad). Primers used for mouse: Ang1 (For 5’-CATTCTTCGCT GCCATTCTG-3’, Rev 5’-TGCAGAGCGTTGGTGTTG TA-3’), and β-actin (For 5’-AACCGTGAAA AGATGACCCAG-3’, Rev 5’-CACAGCCTGGAT GGCTACGTA) (Invitrogen). Gene expression for all samples was normalized to the expression of the house-keeping gene b-actin, which was co-amplified in each reaction. The data for the isolated hepatocytes is presented as log2 fold change of Ang1 KO hepatocytes relative to Ang1 WT hepatocytes, using the average of 2 different isolations of control hepatocytes as baseline expression. The data for the liver tissue is presented as log2 fold change of WT (*n* = 10) and Ang1KO (*n* = 9) mouse livers that were injected with MC-38 cells, using WT livers that were not injected with MC-38 cells as an additional normalization.

### 4.7. Western Blot

Primary hepatocyte cell lysate was prepared in ice-cold RIPA lysis buffer. Protein content was quantitated with the BCA Protein Assay Kit (Thermo Fisher, Waltham, MA, USA, 23227) following the manufacturer’s guide. Protein was resolved by 10% SDS-PAGE gel and then transferred onto PVDF membrane (100V, 2h). After transfer the blots were blocked with 5% non-fat milk dissolved in the TBST buffer for 1h at room temperature with shaking. The Ang1 primary antibody (ab102015, 1:1500, Abcam) and Anti-GAPDH (ab9485, 1:2000, Abcam) were hybridized overnight at 4 °C. The blots were then washed with TBST and then the secondary antibody was added (Rabbit IgG HRP; 170-6515, 1:5000, Bio-Rad) and incubated for one hour at room temperature. The protein blot was then visualized using the Pierce ECL Western Blotting Substrate (32106, Thermo Fisher, Waltham, MA, USA). The analysis of the intensity of the signal was quantitated using ImageLab (Bio-Rad). The values of each band were normalized by dividing them by the value of their GAPDH and normalized to the control sample, which was assigned a value of 1 (Appendix A).

### 4.8. MTT Assay

MC-38 cells were seeded in 96-well plates at a density of 1 × 103 cells/well and cultured in 200 μL DMEM supplemented with 10% FBS. Following overnight starvation in the different hepatocyte conditioned media (see Appendix A), cell viability was assessed using an MTT assay (Abcam) according to the manufacturer’s protocol. Absorbance was measured in an ELISA microplate reader at 562 nm (Infinite M200 pro, Tecan, Mannerdorf, Switzerland). Control conditions consisted of MC-38 cells alone that had an equal volume of serum-free medium. Data are expressed as the mean of a minimum of three independent experiments performed in triplicate.

### 4.9. Proliferation Assay

We performed this assay to determine the effect of Ang1 on proliferation rates in SW620 (human) and MC38 (mouse) colorectal cancer cells. Similar number of cells were cultured in DMEM (#319-005-CL, Wisent Bioproduct, St-Bruno, Qc, Canada.) supplemented with 10% FBS (#085-150, Wisent Inc.), 1× penicillin/streptomycin (450-201-EL, Wisent Inc.) at 37 °C for 24 h. The next day, the media was removed and the cells washed twice with 1X PBS (# 311-010-CL, Wisent Inc.) and fresh serum-free DMEM (#319-005-CL, Wisent Inc.) media was added in the presence or absence of recombinant Ang1 (#130-06, Peprotech, Rocky Hill, NY, USA). The cells were then incubated for another 24 h at 37 °C. Every 24h the cells were collected by trypsinization followed by counting using trypan blue (# 1450021, Bio-Rad) and a hemacytometer.

### 4.10. Scratch Cell Migration Assay

A monolayer of MC-38 cells was grown to 80% confluence in a 6-well plate and at experimental time zero a scratch was made in each well using a pipette tip. The cells were cultured in the different conditioned media (see Appendix A) and imaged at time zero and again 24 h later. A measurement was taken for how much the denuded area had filled in the 24 h period. All experiments were performed in triplicate (*n* = 3).

For the culture of SW620 cells, plates were coated with poly-L-lysine (#A-005-CL, Millipore, Sigma Aldrich) and incubated for 30 min at 37 °C, followed by aspiration and air-drying. The cancer cells were seeded overnight using DMEM (#319-005-CL, Wisent Inc.) supplemented with 10% FBS (#085-150, Wisent Inc.) and 1× penicillin/streptomycin (450-201-EL, Wisent Inc.). The media was aspirated, and a wound was introduced into the monolayer cells using a p200 pipette tip. After washing with PBS (# 311-010-CL, Wisent Inc.), the denuded areas were photographed (0 h). Cells were then cultured using serum-free media at 37 °C in the presence or absence of Ang1 (#130-06, Peprotech). After 24 h, the cells were washed, and the scratched areas were photographed (24 h). All experiments were performed in triplicate (*n* = 3).

### 4.11. Boyden Chamber Invasion Assay

In vitro invasion assay was measured using a Matrigel invasion chamber with 8 µm pores (354483, Thermo Fisher Scientific). MC-38 cells (1 × 10^5^) were cultured in different conditioned media and placed into individual Boyden chamber. The DMEM medium containing 10% FBS was placed in the lower chamber to facilitate chemotaxis. Invasion assays were run for 16 hours, and then cells that passed through the Matrigel membrane were stained with 0.1% crystal violet in 20% methanol. The cells in top chamber were removed by cotton swab. Representative images of migrated and invaded cells were taken by a Nikon Eclipse TS2 microscope (Nikon, Melville, NY, USA).

### 4.12. Statistical Analysis

Statistical analysis was performed with a two-tailed Fisher’s exact test or a two-tailed Student’s *t*-test using GraphPad Prism 6 (GraphPad Software, San Diego, CA, USA) and Microsoft Excel (Microsoft Canada Inc, Mississauga, ON, Canada). Data are presented as a standard error of the mean +/− (SEM). *p*-values of <0.005 were considered significant.

## 5. Conclusions

Overall, these results provide evidence that high expression of the host Ang1 may support vessel co-option formation, tumor growth and metastases and therefore making it a potential target for treatment of these lesions. This study suggests that targeting Ang1 in CRCLM may convert a non-angiogenic lesion (RHGP) into angiogenic lesion (DHGP), which can then be targeted with anti-angiogenic drugs.

## Figures and Tables

**Figure 1 cancers-12-00035-f001:**
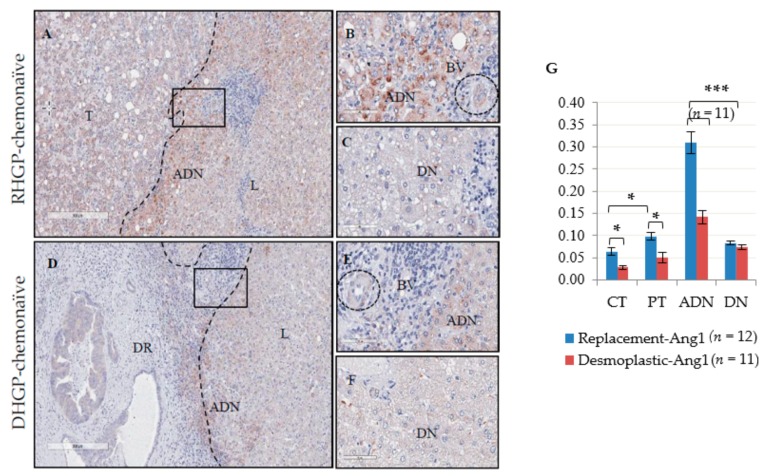
IHC staining of Ang1 in RHGP and DHGP chemonaïve lesions (**A**–**F**). Positivity representing positive pixels staining quantified with Aperio software (**G**). T-Tumor, L-liver, BV-Blood vessel, CT-Center of the tumor, PT-Periphery of the tumor, ADN-Adjacent normal, DN-Distal normal, DHGP ring-DR, dashed-lines represents the border between tumor region and livers data are represented as the mean +/− SEM, significant * *p*-value < 0.005 and *** *p*-value < 0.00005.

**Figure 2 cancers-12-00035-f002:**
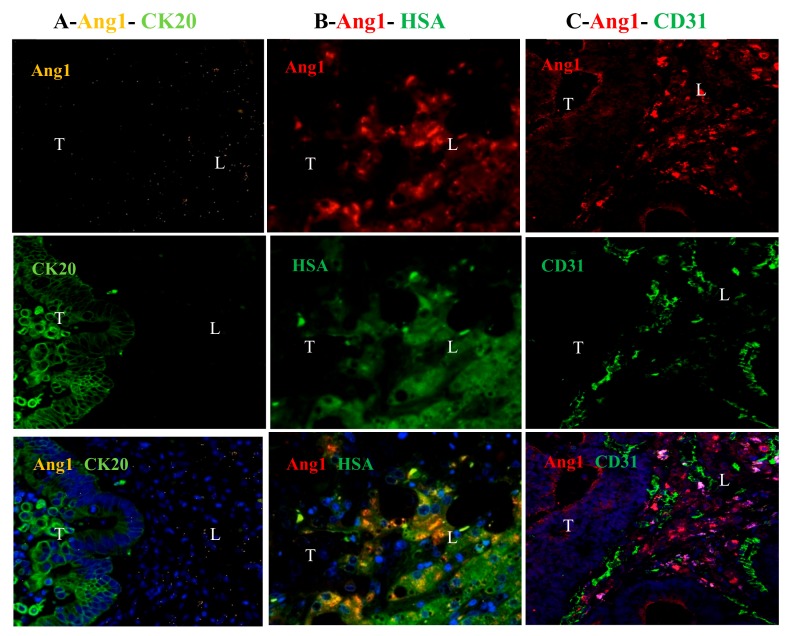
Ang1 was highly expressed in the hepatocytes of the adjacent normal of RHGP chemonaïve lesion (**A**) FISH double staining indicates higher Ang1 mRNA (yellow signal) in hepatocyte region compared to tumor region stained by tumor marker CK20 protein (green signal). Immunofluorescent staining showed (**B**) localization of Ang1 (red signal) and hepatocyte marker HSA (green signal), and (**C**) high expression of Ang1 (red signal) adjacent to vascular marker CD31 (green signal). T-tumor, L-liver.

**Figure 3 cancers-12-00035-f003:**
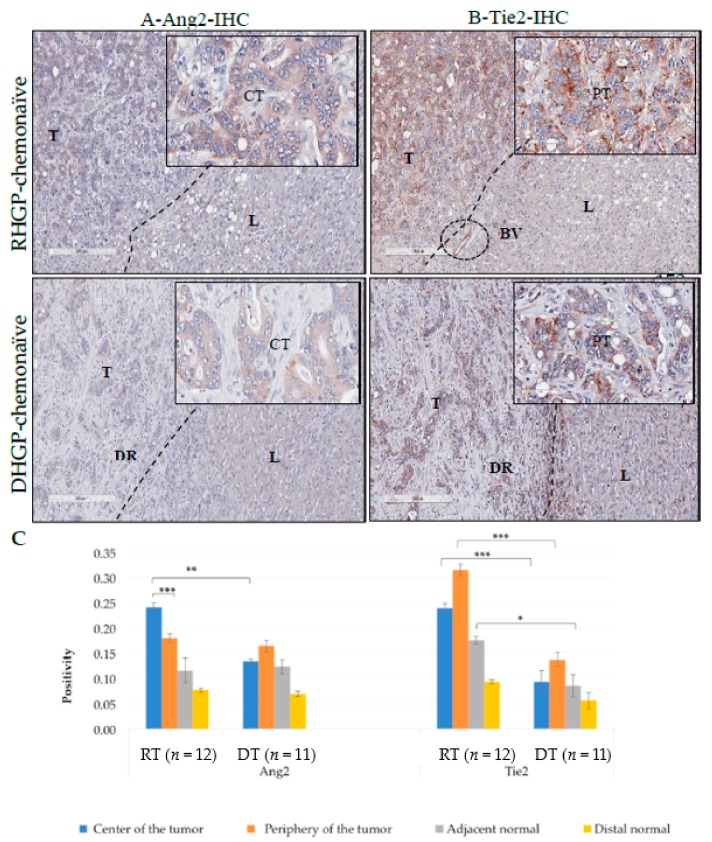
IHC staining of (**A**) Ang2 and (**B**) Tie2 in RHGP and DHGP chemonaïve lesions. (**C**) Positivity representing positive pixels staining quantified with Aperio software. RT- replacement tumor, DT- desmoplastic tumor. T- Tumor, L-liver, BV-Blood vessel, CT-Center of the tumor PT-Periphery of the tumor, DHGP ring-DR, dashed-lines represents the border between tumor region and livers, data are represented as the mean +/− SEM, and * significant *p*-value < 0.005, ** *p*-value < 0.0005 and *** *p*-value < 0.00005.

**Figure 4 cancers-12-00035-f004:**
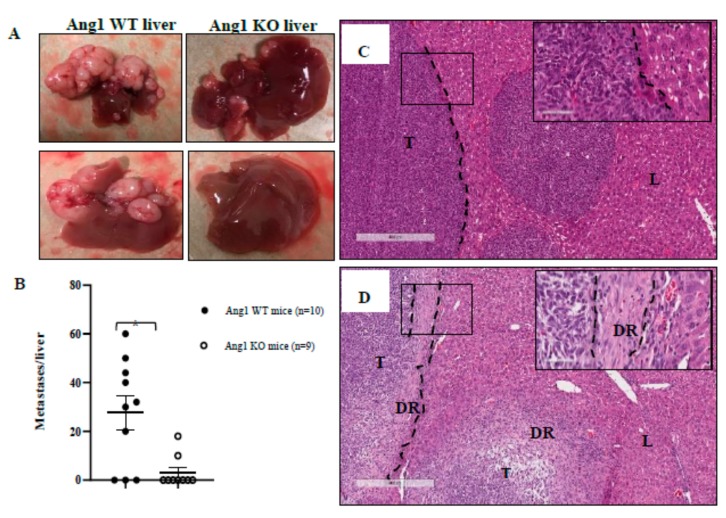
Liver metastatic lesions and H&E staining of metastatic MC-38 mouse colon cancer cells. (**A**) Representative images of control, non-induced mice and Ang1 KO mice that received intra-splenic injections of MC-38 cells. Control mice developed large and multi-focal RHGP lesions whereas the Ang1KO mice had few or no lesions. (**B**) Quantitation of the number of lesions formed in the livers of control vs Ang1 KO mice. A significant difference (*p*-value < 0.005) was observed between the two groups. (**C**) H&E staining of metastatic lesions in control mice demonstrate Replacement HGP phenotype and (**D**) H&E staining of metastatic lesions in Ang1 KO mice demonstrate desmoplastic HGP phenotype. T-Tumor, L-Liver, DR-DHGP ring, dashed-lines represent the border between tumor region and liver, data are represented as the mean +/− SEM, and * significant *p*-value < 0.005.

**Figure 5 cancers-12-00035-f005:**
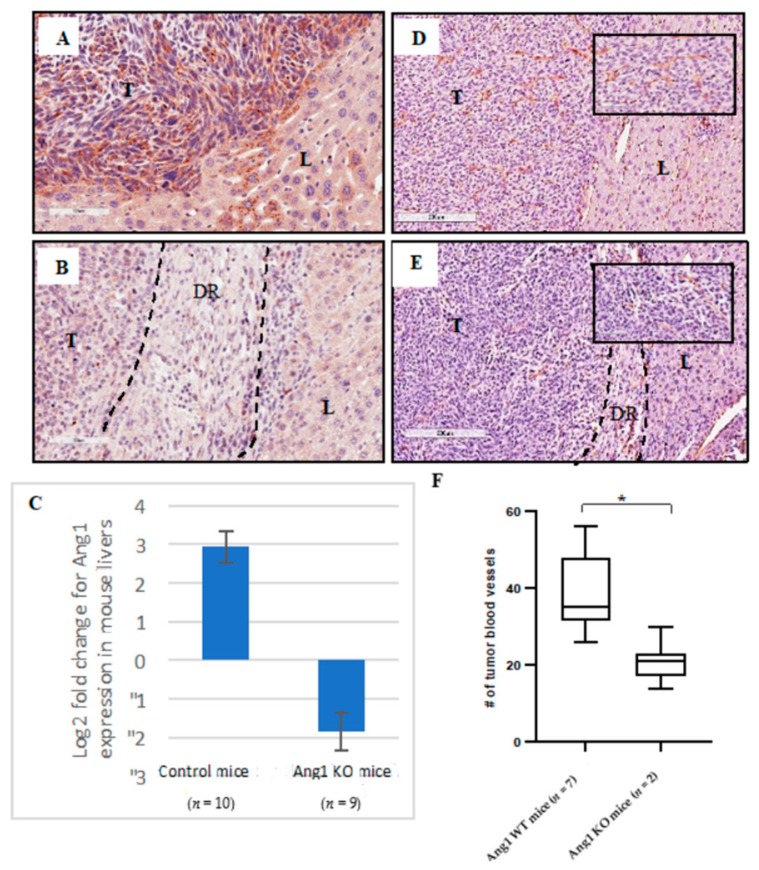
Expression of Ang1 and CD31 in control and Ang1 KO mouse livers. (**A**) IHC staining of Ang1 in control mouse and (**B**) Ang1 KO mouse that developed DHGP tumor. (**C**) qPCR for Ang1 gene expression in control and Ang1 KO mice. qPCR data was first normalized to b-actin and results presented as log2 fold change of control liver tissues with or without tumors for both groups relative to control livers for mice without injection. IHC staining of CD31 in (**D**) control liver and (**E**) Ang1 KO liver. (**F**) Blood vessel counts in control and Ang1 KO mouse tumors. It is important to note that only 2/9 mice developed lesions and therefore the blood vessels counts were performed on the two mice that formed tumors. T-Tumor, L-Liver, DR-DHGP ring, dashed-lines presented the DR the border between tumor region and livers, data are represented as the mean +/− SEM, and * significant *p*-value *<* 0.0005.

**Figure 6 cancers-12-00035-f006:**
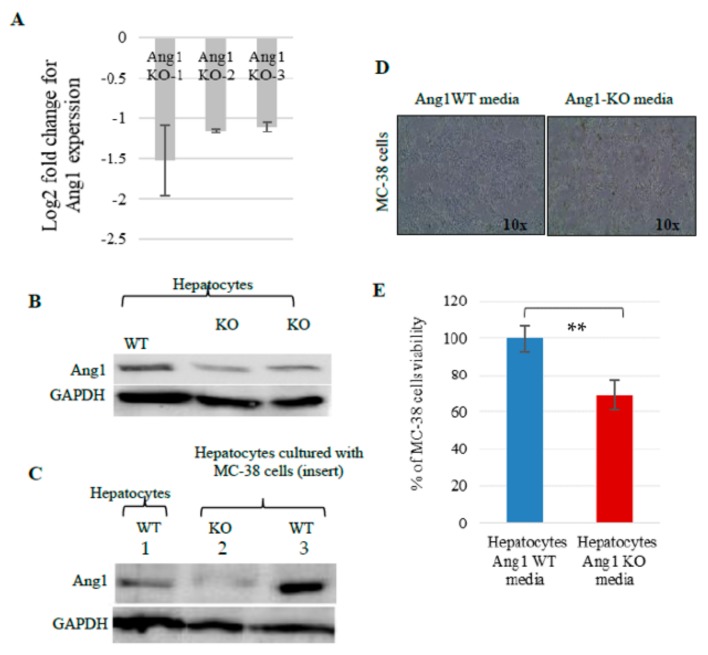
Expression of Ang1 in isolated hepatocytes and MC-38 cell viability. (**A**) qPCR of Ang1 gene expression in isolated Ang1 KO hepatocytes, data is presented as log2 fold change of Ang1 KO hepatocytes relative to Ang1 control hepatocytes. Data was normalized to b-actin. Western blot of Ang1 expression in (**B**) isolated hepatocytes, and (**C**) hepatocytes cultured in serum free medium alone or cultured with MC-38 cells using insert. (**D**) Phase contrast microscopy of MC-38 cells cultured in Ang1 WT and Ang1 KO hepatocyte conditioned media (yellow floating cells represent dead cells). (**E**) MTT assay for MC-38 cells cultured in control or Ang1 KO hepatocyte media. Data are represented as the mean +/− SEM, and ** significant *p*-value < 0.0005.

**Figure 7 cancers-12-00035-f007:**
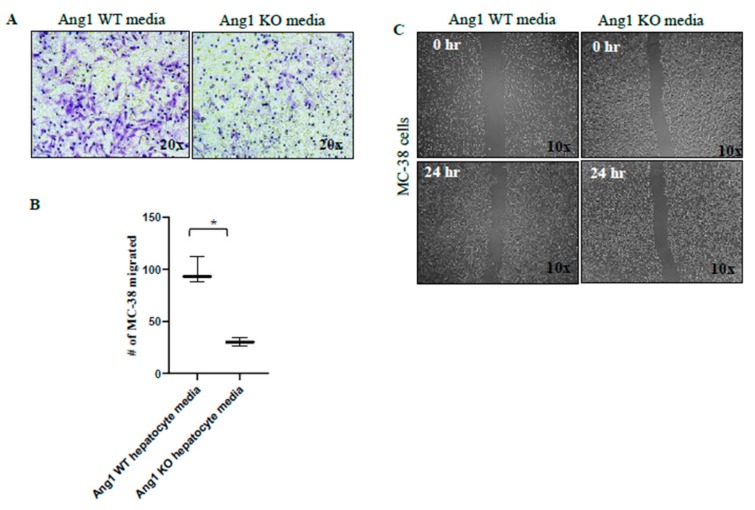
Invasion and Migration assays of MC-38 cells cultured in hepatocytes Ang1 WT and Ang1 KO conditioned media. Ang1 KO hepatocyte media (**A**,**B**) reduced the invasive ability and decreased the number of migrated MC-38 cells through the Matrigel. (**C**) wound healing assay showed inhibition in migration ability of M-38 cells cultured in Ang1 KO hepatocyte media, images from the same area were captured at 0 and 24 h. Data are represented as the mean +/− SEM, and * significant *p*-value < 0.005.

**Figure 8 cancers-12-00035-f008:**
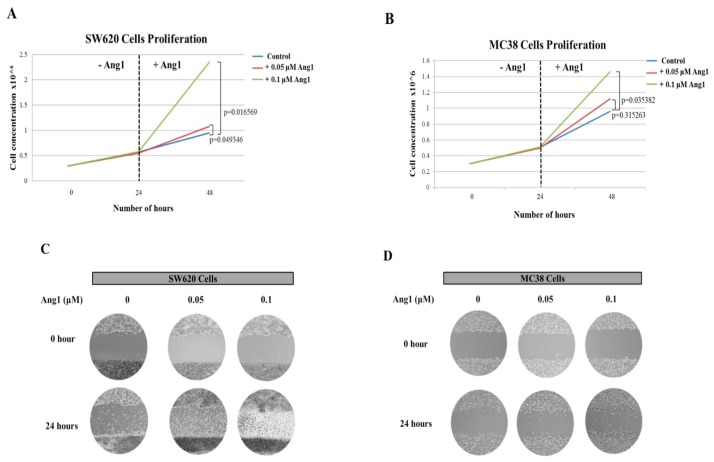
Proliferation and migration assay assessing the effect of recombinant Ang1 on SW620 and MC-38 cells. (**A**) cell proliferation assay of SW620, human colon cancer cells, incubated with vehicle alone (blue line) and 0.1 μM (red line) and 0.05 μM (green line) recombinant Ang1. (**B**) cell proliferation assay of MC-38, mouse colon cancer cells, incubated with vehicle alone (blue line) and 0.1 μM (red line) and 0.05 μM (green line) recombinant Ang1. Scratch assays using (**C**) SW620 and (**D**) MC-38 cells incubated with vehicle alone (0), 0.05 and 0.1 μM of recombinant Ang1. After 24h of culture a wound was generated and pictures taken at time 0 and 24 h after incubation. All experiments were performed in triplicate (*n* = 3).

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
