# Peer review of "Angiopoietin1 Deficiency in Hepatocytes Affects the Growth of Colorectal Cancer Liver Metastases (CRCLM)"

_cancers, 2019, doi:10.3390/cancers12010035_

Round 1
Reviewer 1 Report
Major concerns:
The quality of figures are poor. Especially Figure 6 and 7. Figure legend did not match with representative figure. Intensity of IHC staining is not the quantitative method to quantify Ang1 expression. Other method of quantitation is recommended. Figure 5F, Ang1 KO group animal numbers are relatively less. Tissue Ang1 expression level should be quantified by quantitative method.
Minor concerns:
There are many typos in the text, please review before submission.
For example:
Figure1 legend: ADN
Author Response
Major concerns:
The quality of figures are poor. Especially Figure 6 and 7. Figure legend did not match with representative figure.
We have uploaded higher resolution for all figures. All figure legends have been changed to include more information and Figure legends verified to match figures. Specifically we noticed that Fig 7 was pasted on top of Fig 6, which is why the figure legend did not match. We have corrected this in the revised version of the paper.
Intensity of IHC staining is not the quantitative method to quantify Ang1 expression. Other method of quantitation is recommended.
For the quantitation of Ang1 we used the Aperio ImageScope software. This is recommended by our pathologist and also accepted by the CAP (college of American Pathologist) as a valid method of quantitation. We have published using this method in a reputable journal (Lazaris, A., et al., Vascularization of colorectal carcinoma liver metastasis: insight into stratification of patients for anti-angiogenic therapies. J Pathol Clin Res, 2018.). If this is not acceptable by this journals standards please refer us to another method and we will repeat all the quantitation’s.
Figure 5F, Ang1 KO group animal numbers are relatively less.
The Ang1 KO group animals that we quantitated for tumor vessels are less because only 2 of the 9 KO animals developed lesions. We have included a note in the figure legend to describe this for clarification.
Tissue Ang1 expression level should be quantified by quantitative method.
All Ang1 quantitation is based on positivity scores obtained by using the Aperio Imagescope software. Please see point 2 for response. We remain open to using the method the journal would prefer. For the Western blots we have added a figure in the supplemental showing the quantitation.
Minor concerns:
There are many typos in the text, please review before submission.
We have reviewed the paper and corrected typos we observed (see tracked changes in text).Thank you for your valuable comments
Reviewer 2 Report
Summary
In the manuscript entitled “Angiopoietin1 deficiency in hepatocytes affects the growth of Colorectal Cancer Liver Metastases 3 (CRCLM)” the authors pinpoint an involvement of Ang1 in the context of CRCLM. Particularly, they show the Ang1 levels are higher in hepatocytes adjacent to tumor region of chemo-naïve and treated co-opting of cancers. Interestingly, they found that in Ang1 knockout mice, once received the injection of metastatic colon cancer cells (MC-38), there is a reduction in liver metastases and the development of angiogenic driven desmoplastic liver metastases. In addition, they have performed co-culturing in vitro experiments showing how the down-regulation of Ang1 in hepatocytes could determine the attenuation of colon cancer cells properties (e.g. invasion and migration).
General comment
Overall, the manuscript reports interesting data that can improve the knowledge about CRCLM and Ang1 in the context of angiogenesis and formation of blood vessels. The paper can target a broad audience due to the importance of angiogenesis process, much involved in colon cancer, with the aim to find new therapeutic targets.
Specific comments
The overall structure of the manuscript is very good. The organization of the title, abstract and introduction are quite complete and fully cover the required knowledge for the reader. The language is appropriate as well as the cited bibliography. Generally, the method section is well written and comprehensive/detailed. The results support the conclusion of the manuscript.
However, the manuscript can be improved, regarding the methods and especially presentation of the data.
Following specific suggestions:
Results (lines 177-188): while the text is clear, for the relative figures (figs. 6 and 7) there is some imprecision. Figure 7 legends is stated before the 6; also, it is not very clear which panels correspond to which legend description (e.g. figure 6 legend: panel A is supposed to show the qPCR data, but the only A panel can be seen is the one showing cells). Please, carefully re-check both the figures 6 and 7, the legends and their collocation. Figure 1: this can be improved. The authors should be more precise in the legend. For instance, they should describe the meaning of the dashed-lines (e.g. border between tumoral and liver tissues). Regarding the p-values, even if *** represent p-values < than 0.05, it is good to write the p-value relative of ***. In the panel G, since you are showing a mean, if relevant, state the n value. Figure 2: the authors may be clearer in the figure legend. For example, the panel letters can be moved at the beginning of the relative description, as you have done for the panel A; Figure 3: state the values relative to the significance stars (***); Figure 4: overall, the legend can be written in a clearer way. State the value of ***; Methods/RNA and qPCR section (319): Please, describe more extensively the normalization of the qPCR data (e.g. how the percentage of Ang1 has been calculated). Please, do this in the relevant figures (e.g. figure 5 and more) as well. However, authors may also present the qPCR data as fold-change values (RQ). Methods/RNA and qPCR section (line 313): Regarding the RNA isolation kit (tRNA isolation?), did you mean RNeasy Plus Micro Kit? If so, please correct the text; Methods (line 358): regarding the number of cells in the methods, you report: number of cells 1-1.5x106; did you mean 1-1.5x106? If not correct, please amend this in all the relevant cases; Methods (line 371): specify the Ang1 primary Ab product number/company; Methods/Statistics section (line 398-400): you state that you set the threshold <0.01, however in the figure legends you have made different statements (e.g. figure 4, among the others). Please, check this and be more precise in the relative method section; Bibliography: in the references order, you have missed to mention the references 23 and 24 before the next one (25). So, there is a gap. In particular, the references 23 (line 333) and 24 (line 348) are reported in the methods section. Please, check the overall order of the references and eventually correct the numeration.Author Response
Following specific suggestions:
Results (lines 177-188): while the text is clear, for the relative figures (figs. 6 and 7) there is some imprecision. Figure 7 legends is stated before the 6; also, it is not very clear which panels correspond to which legend description (e.g. figure 6 legend: panel A is supposed to show the qPCR data, but the only A panel can be seen is the one showing cells). Please, carefully re-check both the figures 6 and 7, the legends and their collocation.
We have made the corrections to both Figure 6 and 7. It appears that Figure 7 was superimposed over Figure 6 and this caused confusion. As suggested by Reviewer 1 as well, we have corrected this.
Figure 1: this can be improved. The authors should be more precise in the legend. For instance, they should describe the meaning of the dashed-lines (e.g. border between tumoral and liver tissues). Regarding the p-values, even if *** represent p-values < than 0.05, it is good to write the p-value relative of ***. In the panel G, since you are showing a mean, if relevant, state the n value.
As suggested by the reviewer we have described more details in the legend and have indicated the p value relative to the *. We have also included the n values in panel G.
Figure 2: the authors may be clearer in the figure legend. For example, the panel letters can be moved at the beginning of the relative description, as you have done for the panel A;
We have modified Figure 2 and adjusted the panels to be vertical and indicated in each panel the identity of the protein being stained. We have also modified the description to include more detail.Figure 3: state the values relative to the significance stars (***); Figure 4: overall, the legend can be written in a clearer way. State the value of ***;
We have modified Figure 3 and 4 to include more detail in the legend and indicate the relative p values to the *.Methods/RNA and qPCR section (319): Please, describe more extensively the normalization of the qPCR data (e.g. how the percentage of Ang1 has been calculated). Please, do this in the relevant figures (e.g. figure 5 and more) as well. However, authors may also present the qPCR data as fold-change values (RQ).
We have included in the methods section 4.6 (line 895-912) and in the figure legend of Fig. 5 and 6, more information on how the data was normalized. Also as suggested we have presented the qPCR data a log fold change values.Methods/RNA and qPCR section (line 313): Regarding the RNA isolation kit (tRNA isolation?), did you mean RNeasy Plus Micro Kit? If so, please correct the text;
We have corrected this in the text (line 900).Methods (line 358): regarding the number of cells in the methods, you report: number of cells 1-1.5x106; did you mean 1-1.5x10 ? If not correct, please amend this in all the relevant cases;
We have corrected this in the text (line 724) to 1- 1.5x106.Methods (line 371): specify the Ang1 primary Ab product number/company;
We have specified the Ang1 information in the text (line 918)Methods/Statistics section (line 398-400): you state that you set the threshold <0.01, however in the figure legends you have made different statements (e.g. figure 4, among the others).
Please, check this and be more precise in the relative method section;
We have made changes to the methods section on statistics to include a more precise description ( line 974). We have also made changes to all the figures with more precise indications of the p-values).Bibliography: in the references order, you have missed to mention the references 23 and 24 before the next one (25). So, there is a gap. In particular, the references 23 (line 333) and 24 (line 348) are reported in the methods section. Please, check the overall order of the references and eventually correct the numeration.
We have corrected the order of references in the text and reference list. (lines 1058 - 1070)
Thank you for your valuable comments and input.
Reviewer 3 Report
Major concerns:
Please provide figure 6 and 7 in right order and in accordance with the results section text. Figure 7 lack of key data and description. Please provide the picture in fig.2,3 and 5 in better quality. Please asses the level of VEGEF expression. Please reconsider the change of the first paragraph of introduction section, this part was literally the same as in previous reports published by this group.Author Response
Major concerns:
Please provide figure 6 and 7 in right order and in accordance with the results section text. Figure 7 lack of key data and description.
We have made the corrections to both Figure 6 and 7. It appears that Figure 7 was superimposed over Figure 6 and this caused confusion. As suggested by Reviewer 1 and 2 also, we have corrected this.Please provide the picture in fig.2,3 and 5 in better quality.
We have included higher resolution images for Figures 2, 3 and 5.
Please assess the level of VEGEF expression.
The samples used in this study are serial sections from our previous publication (Lazaris, A., et al., Vascularization of colorectal carcinoma liver metastasis: insight into stratification of patients for anti-angiogenic therapies. J Pathol Clin Res, 2018.). We have included the assessment of VEGEF levels in the results section (line 245).Please reconsider the change of the first paragraph of introduction section, this part was literally the same as in previous reports published by this group.
We have changed the first paragraph of the introduction section so as not to resemble that of our previous papers (lines 50-54 and 58-60).Thank you for your comments to help improve the quality of the paper.
Reviewer 4 Report
The manuscript entitled “Angiopoietin1 Deficiency in Hepatocytes Affects the Growth of Colorectal Cancer Liver Metastasis”, by Ibrahim et al., describes the role of the soluble factor Angiopoietin1 (Ang1) secreted by tumor-adjacent hepatocytes in supporting colorectal tumor cell migration and invasion in vitro and vessel co-option of liver metastases from colorectal tumors in vivo.
The study is interesting and suggests the potential efficacy of targeting Ang1 in patients with colorectal cancer liver metastases. However, there are some aspects that should be improved:
Discussion section shows in general a summary of the results and only includes some explanations to the data. It should be improved to be a real discussion, adding and discussing additional previously reported information and the potential utility of the study for the clinical setting, future work needed etc. For example, is it known whether patients resistant to bevacizumab treatment show lower expression of Ang1 in tumor adjacent tissue? Would it be useful to use Ang1 as predictive biomarker for antiangiogenic therapy? What could be the strategy for a therapeutic inhibition of Ang1 in CRCLM patients? Lines 196-201: Results described here should be further explained/discussed. How do you explain that the conditioned media from direct co-culture of Ang1 KO hepatocytes and CRC cells is different (at least regarding its effect on CRC cell viability) from that obtained from the same co-culture but separated by inserts? In vitro studies could be performed by using culture media supplemented with recombinant human Ang1 (which is commercially available), as an additional experiment to confirm that this secreted protein is the responsible for increasing migration and invasion of tumor cells. Moreover, it is recommended to use a different CRC cell line in order to further confirm the results. Angiopoietin1 is not visible in figure 2A-left, even in the liver area. Please use a better image. Figure 6 is not complete: the figure does not show all the experiments described in the figure legend. Moreover, this figure should be located before Figure 7. Discussion section lines 248-251 describe experimental results (those shown in Figure S6) that have not been previously mentioned in the Results section.
Minor points:
Legend of Figure 7 should be completed to better describe the experiments shown. Letters indicating the different panels are not even mentioned in the legend. Supplementary figures S1 and S2 are only mentioned in the last section (Materials and Methods). Please, reorder supplementary figure numbers or mention them before. Figure S4 does not contain pictures, only the quantitation. Adding pictures would be more informative for readers. Section “2.2 Expression of Ang1 in treated (chemo and chemo plus Bev) CRCLM human samples” includes only supplementary figures; however, Fig. S4 shows Ang1 staining in CRCLM patients treated with chemo +/- bevacizumab, while Fig. 3 (main text) shows expression levels of Ang2 and Tie2, which are not further studied in this work. Authors should consider whether Fig.3 is more important than Fig. S4 to include it in the main text. Lines 113-114 indicate “a significant increase in the Ang1 positivity when comparing the central tumor to the peripheral tumor area of RHGP lesions”. This statistically significant result is not properly represented in figure 1 (with *). Lines 135-136 indicate “we observed high levels of positivity in immune cells in both lesion types (Figure S3)”. However, this Fig. S3 shows only the results in lesions DHGP, not in BOTH lesion types. Line 245: “Ang1 in hepatocytes is directly correlated with the presence of tumor cells”. There is no direct correlation, as there is only a picture but it is not quantified. Line 400: “p-values of <0.01 were considered significant”. A threshold of p-value <0.05 is indicated in most figure legends. Exponential numbers are not correctly written with superscripts (lines 332, 358, 377, 391, etc.) Graphs in Figures 4B and 7B are too small. There are some typos along the manuscript: legend of Fig1 should show “ADN” instead of “AND” as well as “mean” instead of the symbols shown; the title 2.2 should be separated from the text of section 2.1, and the following sections should be numbered consecutively; legend of Fig. 3 includes the abbreviations of Adjacent normal and Distal normal, but they are not used in the figure; some symbols are missing in the figures (DN in Fig.1C, L in Fig. 3A-right, L in Fig. 5A-right), etc.. Line 313: “tRNA” should be replaced by “Total RNA”.
Author Response
Discussion section shows in general a summary of the results and only includes some explanations to the data. It should be improved to be a real discussion, adding and discussing additional previously reported information and the potential utility of the study for the clinical setting, future work needed etc. For example, is it known whether patients resistant to bevacizumab treatment show lower expression of Ang1 in tumor adjacent tissue? Would it be useful to use Ang1 as predictive biomarker for antiangiogenic therapy? What could be the strategy for a therapeutic inhibition of Ang1 in CRCLM patients?
We have modified the conclusion to include additional explanations and discussion on the implications of our findings in relation to other studies. (lines 502-519, 539-617). We discuss how and if Ang1 can be used as a predictive marker and therapeutic strategies for patients with RHGP lesions.
Lines 196- 201: Results described here should be further explained/discussed. How do you explain that the conditioned media from direct co-culture of Ang1 KO hepatocytes and CRC cells is different (at least regarding its effect on CRC cell viability) from that obtained from the same co-culture but separated by inserts? In vitro studies could be performed by using culture media supplemented with recombinant human Ang1 (which is commercially available), as an additional experiment to confirm that this secreted protein is the responsible for increasing migration and invasion of tumor cells.
Moreover, it is recommended to use a different CRC cell line in order to further confirm the results.
We have added additional explanations to this section in the results and have included additional data were we added recombinant human Ang1 to two different colon cancer cell lines and demonstrate an increase in migration and cell growth introduction: lines 115-117, results new section 2.5 (lines 466-475), discussion lines 592-595 and Figure 8Angiopoietin1 is not visible in figure 2A-left, even in the liver area. Please use a better image.
Figure 6 is not complete: the figure does not show all the experiments described in the figure legend. Moreover, this figure should be located before Figure 7.
We have added a higher resolution of Fig 2A and made changes to Figure 6 and 7 such that the legends have more details and describe the figures. Also please note that Fig 7 was superimposed on Fig 6 during the conversion which could have added to the confusion, nevertheless we have made the changes suggested.Discussion section lines 248- 251 describe experimental results (those shown in Figure S6) that have not been previously mentioned in the Results section.
We have kept Figure S6, which as been renumbered to Figure S5 in the edited version of the paper in the discussion as these are preliminary findings with small number of animals and we have not stained yet for Ang1 and therefore prefer not to include in the results section.
Minor points:
Legend of Figure 7 should be completed to better describe the experiments shown. Letters indicating the different panels are not even mentioned in the legend.
We have made the modifications suggested in the legend to Figure 7.Supplementary figures S1 and S2 are only mentioned in the last section (Materials and Methods). Please, reorder supplementary figure numbers or mention them before.
We have reordered the supplementary figures to reflect the order in the paper.Figure S4 does not contain pictures, only the quantitation. Adding pictures would be more informative for readers. Section “2.2 Expression of Ang1 in treated (chemo and chemo plus Bev) CRCLM human samples” includes only supplementary figures; however, Fig. S4 shows Ang1 staining in CRCLM patients treated with chemo +/- bevacizumab, while Fig. 3 (main text) shows expression levels of Ang2 and Tie2, which are not further studied in this work. Authors should consider whether Fig.3 is more important than Fig. S4 to include it in the main text.
Since the focus of the paper is on the discovery of Ang1, focusing on chemonaïve patients we feel that including the treated samples in the main paper will take away from the message. Furthermore, we cannot compare the treated RHGP lesions to the DHGP lesions as treatment causes a response in the DHGP lesions and there is less than 5-10% of viable cells left. Therefore we feel this is not a fair comparison. The Ang2 and Tie2 is more relevant, as we note in the discussion (lines 502-519), to further support our hypothesis of vessel co-option.Lines 113-114 indicate “a significant increase in the Ang1 positivity when comparing the central tumor to the peripheral tumor area of RHGP lesions”. This statistically significant result is not properly represented in figure 1 (with *).
We have indicated in the figure legend the proper p-value.
Lines 135-136 indicate “we observed high levels of positivity in immune cells in both lesion types (Figure S3)”. However, this Fig. S3 shows only the results in lesions DHGP, not in BOTH lesion types.
The RHGP lesions have very few immune cells and the purpose of figure S3 is to demonstrate the expression of Tie2 on immune cells. Figure 3 panel B shows the RHGP lesions with positive staining in the peripheral region and there are small positive immune cells present. We are working on another paper describing the subtype of T cells that are positive in both DHGP and RHGP.
Line 245: “Ang1 in hepatocytes is directly correlated with the presence of tumor cells”. There is no direct correlation, as there is only a picture but it is not quantified.
We have modified the statement in the paper to say “increase in Ang1 expressing hepatocytes with increasing amounts of tumor cells”. (line 272). We removed the statement direct correlation and kept it as a pure observational comment.Line 400: “p-values of <0.01 were considered significant”. A threshold of p-value <0.05 is indicated in most figure legends.
Exponential numbers are not correctly written with superscripts (lines 332, 358, 377, 391, etc.)
Graphs in Figures 4B and 7B are too small.
We have corrected the p-value in the text and figure legends. The superscripts have been corrected and the graphs in figure 4 B and 7B enlarged.
There are some typos along the manuscript: legend of Fig1 should show “ADN” instead of “AND” as well as “mean” instead of the symbols shown; the title 2.2 should be separated from the text of section 2.1, and the following sections should be numbered consecutively; legend of Fig. 3 includes the abbreviations of Adjacent normal and Distal normal, but they are not used in the figure; some symbols are missing in the figures (DN in Fig.1C, L in Fig. 3A-right, L in Fig. 5A-right), etc.. Line 313: “tRNA” should be replaced by “Total RNA”.
We have made corrections to all the comments suggested and corrected typos in the text and legends.Thank you for your valuable comments and suggestions for improvements to the paper.
Round 2
Reviewer 3 Report
All suggestions were incorporated. The manuscript is now suitable for publication in Cancers.
Author Response
The reviewer did not have any additional comments to address.
They only indicate that it is suitable for publication.
We would like to thank them for their time and effort in reviewer our paper and their valuable comments.
Reviewer 4 Report
Scientifically, the authors have satisfactorily addressed all my comments. However, there are still some aspects that should be improved:
There are still some typos along the text. Figures 4 and 7 appear twice. Figure 5 needs to be formatted. Some areas are overlapped. Figure 6: “D” is missing in the figure itself (it is mentioned in the legend). Figure 1: The indicated n of desmoplastic and replacement lesions (11/11) does not fit with that in the text (line 108: 11/12).Author Response
There are still some typos along the text.
We have gone through the text with spell check and corrected a few typos.
Figures 4 and 7 appear twice.
The version received by the editor only has one figure 4 and 7. The figure legends were distorted and we corrected this.
Figure 5 needs to be formatted. Some areas are overlapped.
Figure 5 has been reformatted to fix the overlaps.
Figure 6: “D” is missing in the figure itself (it is mentioned in the legend).
We have added D which was omitted.
Figure 1: The indicated n of desmoplastic and replacement lesions (11/11) does not fit with that in the text (line 108: 11/12).
This is a typo. We have corrected Figure 1 to show n=12 for the replacement lesions.